# Elliptic deformation of $\mathcal{W}_N$-algebras

**Jean Avan[1], Luc Frappat[2]⋆ and Eric Ragoucy[2]**

**1** Laboratoire de Physique Théorique et Modélisation (CNRS UMR 8089),
Université de Cergy-Pontoise, F-95302 Cergy-Pontoise, France
**2** Laboratoire d'Annecy-le-Vieux de Physique Théorique LAPTh,
Université Grenoble Alpes, USMB, CNRS, F-74000 Annecy

⋆ luc.frappat@lapth.cnrs.fr

## Abstract

We construct $q$-deformations of quantum $\mathcal{W}_N$ algebras with elliptic structure functions. Their spin $k+1$ generators are built from $2k$ products of the Lax matrix generators of $\mathcal{A}_{q,p}(\widehat{gl}(N)_c))$. The closure of the algebras is insured by a critical surface condition relating the parameters $p, q$ and the central charge $c$. Further abelianity conditions are determined, either as $c = -N$ or as a second condition on $p, q, c$. When abelianity is achieved, a Poisson bracket can be defined, that we determine explicitly. One connects these structures with previously built classical $q$-deformed $\mathcal{W}_N$ algebras and quantum $\mathcal{W}_{q,p}(\mathfrak{sl}_N)$.

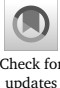

# 1   Introduction

Deformations of $\mathcal{W}_N$ algebras, generalizing the notion of $q$-deformed Virasoro algebras to higher spin structures, were first proposed as classical Poisson structures [1]. Quantization of these structures as $\mathcal{W}_{q,p}(A_N)$ algebras was achieved in [2, 3], together with a realization by free fields. They exhibit remarkable $\Theta$-functions exchange factors and are identified as relevant structures in a number of contexts [4]. Much interest has been recently devoted to their occurence in extensions of the AGTW conjecture to 5D $SU(N)$ gauge theories [5]; in quantum $q$-Langlands correspondence [6]; in the computation of form factors for affine Toda $A_{N-1}^{(1)}$ models [7]; in connection with Macdonald polynomials [3, 8].

   We had proposed time ago [9, 10] several approaches to identify these structures from generalizations of our construction in the Deformed Virasoro Algebra (DVA) case, essentially using as building blocks quadratic expressions in terms of the Lax representation of elliptic quantum algebras [11, 12] (this approach originates in the work [13]). Such constructions appeared natural, given the number of deformation parameters and the occurence of elliptic functions as structure functions for the DVA, then for the $q$-deformed $\mathcal{W}_N$ algebras. However this construction suffered from the defect that it depended on the explicit introduction of some permutation operators $P_{ij}$ acting on pairs of loop-representation spaces $V_i \otimes \mathbb{C}(\lambda_i) \otimes V_j \otimes \mathbb{C}(\lambda_j)$, hence not well-defined at the representation level.

   This difficulty can now be overtaken due to recent advances in understanding the structure of higher spin operators in quantum affine algebras, see Sugawara construction in [14], and quantum determinant for elliptic quantum algebras of vertex type [15].

   We will therefore propose here a consistent construction of generators for $q$-deformations of $\mathcal{W}_N$ algebras with elliptic structure functions hereafter denoted $q$-$\mathcal{W}_N$ . We start from the elliptic quantum algebra $\mathcal{A}_{q,p}(\widehat{gl}(N)_c)$ [12] expressed as exchange relations for abstract generators encapsulated in related Lax matrices $L$. The $q$-$\mathcal{W}_N$ generators are then built as traces of products of $k$ generators of $L$ type with $k$ generators of $L^{-1}$ type. This is in contrast with the previous construction in [9] which used products of $k$ bilinears $L.L^{-1}$. The $2k$ matrices are intertwined by an operator identified to the antisymmetrizer on $(\mathbb{C}^N)^{\otimes k}$, hence these generators are now unambiguously defined. As was the case in DVA [16], we identify 'critical' surfaces $\mathscr{S}$ characterized by algebraic relations between $p, q$ (elliptic nome and deformation parameter of $\mathcal{A}_{q,p}(\widehat{gl}(N)_c)$) and the central charge $c$. On surfaces $\mathscr{S}$ the exchange relations between the generating functions close, leading to a $q$-$\mathcal{W}_N$ algebra. A further critical condition on $p, q, c$ sends the structure functions to one, the algebra becomes thus abelian and a well-defined Poisson structure is then defined. Connections with the original objects in [1–3] will be commented upon in the course of the paper.

   The paper is organized as follows. We introduce in section 2 the basic notations and definitions of higher rank elliptic algebras $\mathcal{A}_{q,p}(\widehat{gl}(N)_c)$. Technical details are supplied in Appendices A and B. We state in section 3 the major result of our paper, giving an explicit construction of $q$-deformed $\mathcal{W}_N$ quantum algebras as subalgebras of $\mathcal{A}_{q,p}(\widehat{gl}(N)_c)$ for "critical conditions" on $p, q, c$ labelled by two integers $m, n \in \mathbb{Z}$. Exchange functions differ from the original ones

in [2] but coincide with those in [9]. The $\mathcal{W}_{q,p}(A_N)$ exchange functions of [2] are recovered by introducing a modified elliptic algebra.

We proceed in section 4 with identification of the second "abelianity condition" yielding now commutation of the quantum $q$-$\mathcal{W}_N$ generators and allowing a computation of the Poisson structure thus generated, similar to the ones of [9,16]. In the case $m = -n = 1$, the critical surface degenerates to $c = -N$ (without further relation between $q$ and $p$) and yields an extended center for the algebra. Computation of the Poisson bracket is technically more intricate and will be explicitly given in section 5. A comparison with the Poisson algebra derived in [1] is established and discussed .

The more technical aspects of the proofs in section 3 are then given in section 6. We finally comment on some open problems in section 7.

## 2 The elliptic quantum algebra $\mathcal{A}_{q,p}(\widehat{gl}(N)_c)$

We recall here the definition of the elliptic quantum algebra $\mathcal{A}_{q,p}(\widehat{gl}(N)_c)$ [9, 11, 12]. We consider the free associative algebra generated by the operators $L_{ij}[n]$ where $i, j \in \mathbb{Z}_N$ and $n \in \mathbb{Z}$, and we define the formal series

$$L_{ij}(z) = \sum_{n \in \mathbb{Z}} L_{ij}[n] z^n \tag{2.1}$$

encapsulated into a $N \times N$ matrix

$$L(z) = \begin{pmatrix} L_{11}(z) & \cdots & L_{1N}(z) \\ \vdots & & \vdots \\ L_{N1}(z) & \cdots & L_{NN}(z) \end{pmatrix}. \tag{2.2}$$

We adjoin an invertible central element written as $q^c$ ($q$ is the deformation parameter and $c$ the central charge).

One defines $\mathcal{A}_{q,p}(\widehat{gl}(N)_c)$ by imposing the following constraints on $L_{ij}(z)$:

$$\widehat{R}_{12}(z/w) L_1(z) L_2(w) = L_2(w) L_1(z) \widehat{R}^*_{12}(z/w), \tag{2.3}$$

where $L_1(z) \equiv L(z) \otimes \mathbb{I}$, $L_2(z) \equiv \mathbb{I} \otimes L(z)$, $\widehat{R}_{12}(z) \equiv \widehat{R}_{12}(z, q, p)$ is the $N$-elliptic $R$-matrix [17–19] defined in Appendix B and $\widehat{R}^*_{12}(z) = \widehat{R}_{12}(z, q, p^* = pq^{-2c})$, see (B.14).

It is useful to introduce the following two matrices:

$$L^+(z) = L(q^{\frac{c}{2}} z), \tag{2.4}$$

$$L^-(z) = (g^{\frac{1}{2}} h g^{\frac{1}{2}}) L(-p^{\frac{1}{2}} z) (g^{\frac{1}{2}} h g^{\frac{1}{2}})^{-1}, \tag{2.5}$$

where the matrices $g$ and $h$ are defined in (B.4) and (B.7). They obey coupled exchange relations following from (2.3), periodicity condition (B.13) and unitarity property (B.17) for the matrices $\widehat{R}_{12}$ and $\widehat{R}^*_{12}$:

$$\begin{aligned} \widehat{R}_{12}(z/w) L^\pm_1(z) L^\pm_2(w) &= L^\pm_2(w) L^\pm_1(z) \widehat{R}^*_{12}(z/w), \\ \widehat{R}_{12}(q^{\frac{c}{2}} z/w) L^+_1(z) L^-_2(w) &= L^-_2(w) L^+_1(z) \widehat{R}^*_{12}(q^{-\frac{c}{2}} z/w). \end{aligned} \tag{2.6}$$

These RLL relations have a similar form to that used to define the quantum affine algebras $\mathcal{U}_q(\widehat{gl}_N)$ (although here $L^+(z)$ and $L^-(z)$ are not independent Lax matrices) and are therefore useful to study the non-elliptic limit $p \to 0$.

# 3 Deformed $\mathcal{W}$-algebras in $\mathcal{A}_{q,p}(\widehat{gl}(N)_c)$

## 3.1 Construction of the quantum $W$ generators

**Theorem 3.1**
*In $\mathcal{A}_{q,p}(\widehat{gl}(N)_c)$, for $1 \leq k \leq N$, we introduce the generators*

$$t_{m,n}^{(k)}(z) = \mathrm{tr}_{1...k}\left(\mathbb{M}\prod_{i=1}^{\overleftarrow{k}}L_i\big((-p^{*\frac{1}{2}})^n z_i\big)\tilde{\mathbb{M}}\prod_{i=1}^{\overrightarrow{k}}L_i(z_i)^{-1}A_k^{(N)}\right), \tag{3.1}$$

*where $z_i = q^{i-1-(k-1)/2}z$, $A_1^{(N)} = \mathbb{I}_N$ and $A_k^{(N)}$, $k \geq 2$, is the antisymmetrizer on $(\mathbb{C}^N)^{\otimes k}$. The matrices $\mathbb{M}$ and $\tilde{\mathbb{M}}$ are constructed from single-space objects*

$$M = (g^{\frac{1}{2}}hg^{\frac{1}{2}})^{-m} \text{ and } \tilde{M} = (g^{\frac{1}{2}}hg^{\frac{1}{2}})^{-n}, \quad m,n \in \mathbb{Z} \tag{3.2}$$

*as*

$$\mathbb{M} = \prod_{i=1}^{k}M_i \quad and \quad \tilde{\mathbb{M}} = \prod_{i=1}^{k}\tilde{M}_i,$$

*where $M_i$ (resp. $\tilde{M}_i$) is the matrix $M$ (resp. $\tilde{M}$) acting in space $i$.*
    *They obey the following exchange relation:*

$$t_{m,n}^{(k)}(z)\,L(w) = \prod_{i=1}^{k}\frac{\mathcal{F}_{-m}(z_i/w)}{\mathcal{F}_n^*(z_i/w)}\,L(w)\,t_{m,n}^{(k)}(z), \tag{3.3}$$

*provided the constraint equation ("surface condition" $\mathscr{S}_{m,n}$)*

$$(-p^{\frac{1}{2}})^m(-p^{*\frac{1}{2}})^n = q^{-N} \tag{3.4}$$

*holds in the three-dimensional parameter space spanned by $q, p, c$.*
    *The function $\mathcal{F}_a(x)$ is expressed in terms of the function $\mathcal{U}(x)$ defined in (B.18) as:*

$$\mathcal{F}_a(x) = \begin{cases} \displaystyle\prod_{\ell=0}^{a-1}\mathcal{U}\big((-p^{\frac{1}{2}})^\ell x\big) & \text{for } a > 0 \\ 1 & \text{for } a = 0 \\ \displaystyle\prod_{\ell=1}^{|a|}\mathcal{U}\big((-p^{\frac{1}{2}})^{-\ell}x\big)^{-1} & \text{for } a < 0 \end{cases} \quad ; \quad \mathcal{F}_a^*(x) = \mathcal{F}_a(x)\big|_{p \to p^*}. \tag{3.5}$$

**Proof:** The proof is postponed to section 6. ∎

Note that the generator $t_{m,n}^{(1)}(z)$ corresponds to the deformed Virasoro algebra introduced in [16], and obeying on the surface $\mathscr{S}_{m,n}$ the relations (3.3) for $k = 1$.

**Remark 3.1** When $k = N$, one can compute directly that

$$t_{m,n}^{(N)}(z) = \det(M)\det(\tilde{M})\,\mathrm{qdet}\big((-p^{*\frac{1}{2}})^n q^{N-1}z\big)\Big(\mathrm{qdet}(q^{N-1}z)\Big)^{-1}, \tag{3.6}$$

where $\mathrm{qdet}(z)$ is the quantum determinant [15] defined by

$$L_1(z)L_2(z/q)\cdots L_N(zq^{1-N})A_N^{(N)} = A_N^{(N)}\,\mathrm{qdet}(z).$$

We have also used $\text{tr}_{1...N}\left(\mathbb{M}A_N^{(N)}\right) = \det(M)$.

Relation (3.6) is a direct consequence of the definition of the quantum determinant, and is valid without any surface condition. Since the quantum determinant is central in $\mathcal{A}_{q,p}(\widehat{gl}(N)_c)$, it implies that $t_{m,0}^{(N)}(z)$ commutes with $L(w)$. Indeed, one can check that the structure function in (3.3) is exactly 1 for $k = N$. In fact, it is easy to see that $\prod_{i=1}^{N}\mathcal{U}(q^i x) = 1$, $\forall x$ because of the property (A.9).

**Remark 3.2** If $n = 0$, the $L$ matrices simplify, and we get $t_{m,0}^{(k)}(z) = \text{tr}_{1...k}\left(\mathbb{M}A_k^{(N)}\right)$, which commutes with $L(w)$. Thus, we should find that for $n = 0$ the structure function in (3.3) is 1 whenever $t_{m,0}^{(k)}(z) \neq 0$. This scalar object is identified with the sum of principal minors of order $k$ for $\mathbb{M}$ which is also the symmetric polynomial of order $k$ in the eigenvalues of $\mathbb{M}$. It is easy to see that either $mk$ is not proportional to $N$, and then $t_{m,0}^{(k)}(z) = 0$, or $mk = aN$ for some integer $a$, in which case $t_{m,0}^{(k)}(z) \neq 0$.

In the latter case, either $k = N$ and then we are back to remark 3.1, or $k < N$ and then $m$ and $N$ are not coprime and have a common divisor. Solving now the surface equation $(-p^{\frac{1}{2}})^m = q^{-N}$, redefining $m, N$ by expliciting their common divisor and coprime factors, and suitably reordering the products in the structure function of (3.3), allows to directly show, using again property (A.9), that this structure function simplifes to 1. Hence (3.3) still applies on the surface $\mathscr{S}_{m,0}$.

Each choice of a pair of integers $(m, n)$ characterizes a surface $\mathscr{S}_{m,n}$. On this surface $\mathscr{S}_{m,n}$, one can then define a quadratic algebra generated by $t_{m,n}^{(k)}(z)$ with $1 \leq k \leq N$ and exchange relations given by the following corollary:

**Corollary 3.2**
*On the surface $\mathscr{S}_{m,n}$ and for $1 \leq k, k' \leq N$, we have the following quadratic exchange relations:*

$$t_{m,n}^{(k)}(z)\, t_{m,n}^{(k')}(w) = \prod_{i=(1-k)/2}^{(k-1)/2}\prod_{j=(1-k')/2}^{(k'-1)/2}\mathcal{Y}_{m,n}(q^{i-j}z/w)\, t_{m,n}^{(k')}(w)\, t_{m,n}^{(k)}(z), \qquad (3.7)$$

*where the function $\mathcal{Y}_{m,n}(x)$ is given by*

$$\mathcal{Y}_{m,n}(x) = \frac{\mathcal{F}_n^*(x)\mathcal{F}_m((-p^{*\frac{1}{2}})^n x)}{\mathcal{F}_n^*((-p^{*\frac{1}{2}})^{-n}x)\mathcal{F}_m(x)} = \frac{\mathcal{F}_n^*(x)\mathcal{F}_{-n}^*(x)}{\mathcal{F}_m(x)\mathcal{F}_{-m}(x)}. \qquad (3.8)$$

**Proof:** This is a direct consequence of the definition (3.1) and the relation (3.3). ∎
Note again that one recovers at $k = k' = 1$ the DVA case with its exchange function $\mathcal{Y}_{m,n}(x)$ [16].

## 3.2 Comparison with the original quantum $W$ generators

The original quadratic exchange relations between generators $t^{(k)}(z)$ and $t^{(k')}(w)$ were derived in [2]. One easily convinces oneself that there is no surface $\mathscr{S}_{m,n}$ that would lead to these exact quadratic exchange relations for any choice of surface indices $m, n$ in (3.7). Therefore we cannot identify any set $t_{m,n}^{(k)}(z)$ from (3.1) with the generators $t^{(k)}(z)$ in [2].
However, as in the DVA case [16], a contact between the general approach defined in the present paper and the results of [2] can be done using a slightly modified algebra. Indeed, let us consider the algebra defined by Lax matrices $\mathsf{L}^{\pm}(z)$ subject to the RLL relations

$$R_{12}(z/w)\,\mathsf{L}_1^{\pm}(z)\,\mathsf{L}_2^{\pm}(w) = \mathsf{L}_2^{\pm}(w)\,\mathsf{L}_1^{\pm}(z)R_{12}^*(z/w),$$
$$R_{12}(q^{\frac{c}{2}}z/w)\,\mathsf{L}_1^+(z)\,\mathsf{L}_2^-(w) = \mathsf{L}_2^-(w)\,\mathsf{L}_1^+(z)R_{12}^*(q^{-\frac{c}{2}}z/w), \qquad (3.9)$$

where $R_{12}(x)$ is the *unitary R-matrix* of $\mathcal{A}_{q,p}(\widehat{gl}(N)_c)$ defined by (B.3) and the Lax matrices $L^\pm(z)$ are considered as *independent* (see remark 3.3 below).

The construction presented in Theorem 3.1 can be repeated using this algebra, introducing generators $t_{m,n}^{(k)}(z)$ given by a formula analogous to (3.1) but now expressed in terms of $L^\pm(z)$:

$$t_{m,n}^{(k)}(z) = \mathrm{tr}_{1\dots k}\Big(\overset{\leftarrow}{\prod_{i=1}^{k}}(g^{\frac{1}{2}}hg^{\frac{1}{2}})_i^{-m+1}\, L_i^+\big(q^{c/2}z_i\big)(g^{\frac{1}{2}}hg^{\frac{1}{2}})_i^{-n-1}\overset{\rightarrow}{\prod_{i=1}^{k}}L_i^-(z_i)^{-1}\,A_k^{(N)}\Big). \qquad (3.10)$$

Since the RLL relations (3.9) differ only by normalization factors from (2.6), see eq. (B.14), the surface on which $t_{m,n}^{(k)}(z)$ closes quadratically is unchanged. Focusing on the case $m=2$ and $n=-1$, on the surface $\mathscr{S}_{2,-1}$, one gets

$$t_{2,-1}^{(k)}(z)\,t_{2,-1}^{(k')}(w) = \prod_{i=(1-k)/2}^{(k-1)/2}\prod_{j=(1-k')/2}^{(k'-1)/2} Y_{2,-1}(q^{i-j}z/w)\, t_{2,-1}^{(k')}(w)\,t_{2,-1}^{(k)}(z), \qquad (3.11)$$

where

$$Y_{2,-1}(x) = \frac{\Theta_{q^{2N}}(x^{-2})\,\Theta_{q^{2N}}(q^2x^{-2})\,\Theta_{q^{2N}}(q^{2+2c}x^2)\,\Theta_{q^{2N}}(q^{-2c}x^2)}{\Theta_{q^{2N}}(x^2)\,\Theta_{q^{2N}}(q^2x^2)\,\Theta_{q^{2N}}(q^{-2c}x^{-2})\,\Theta_{q^{2N}}(q^{2+2c}x^{-2})}. \qquad (3.12)$$

One recognizes in (3.12) the result of [2] with the redefinition $(q,p)$ in [2] $\to (p, q^2)$ here. It is therefore consistent to identify our generators $t_{2,-1}^{(k)}(z)$ with the $t^{(k)}(z)$ in [2].

**Remark 3.3** Note that the construction presented here is based on a modification of the normalization of the RLL relations, leading to a modification of the structure functions $\mathcal{Y}_{m,n}(x)\to Y_{m,n}(x)$. This can only be achieved using $L^\pm(z)$ generators instead of original $L(z)$ ones: the RLL relations (2.3) are insensitive to a change of normalization of the $R$-matrices by a $p$-independent function, in contrast to (3.9). Hence, this implies the definition of an alternative quantum elliptic algebra, in which the generators $L^\pm(z)$ are *a priori* kept independent. In that case, this algebra seems to exhibit twice many generators as $\mathcal{A}_{q,p}(\widehat{gl}(N)_c)$. One may wonder whether a relation between $L^+(z)$ and $L^-(z)$ exists, necessarily distinct from (2.4)–(2.5), bringing this algebra in a form closer to $\mathcal{A}_{q,p}(\widehat{gl}(N)_c)$. This issue and the precise nature of this alternative elliptic quantum algebra is beyond the scope of the paper.

## 4 Abelian quadratic subalgebras and Poisson structures

### 4.1 Abelian quadratic subalgebras

Since the exchange function in corollary 3.2 is a product of the $\mathcal{Y}_{m,n}$ function with different arguments, the conditions under which the exchange relations (3.7) become abelian are the same as in the DVA case. We just recall them here, see [16] for more details.

**Proposition 4.1 (Abelian subalgebras in $\mathcal{A}_{q,p}(\widehat{gl}(N)_c)$)**
*On the surface $\mathscr{S}_{m,n}$, the generators $t_{m,n}(z)$ realize an abelian subalgebra in $\mathcal{A}_{q,p}(\widehat{gl}(N)_c)$ when one of the following conditions is satisfied:*

- *for $|m|, |n| > 1$:*

$$c = \frac{N}{nm}\big(\lambda'm - \lambda n\big), \quad -p^{\frac{1}{2}} = q^{-N\lambda/m}, \quad -p^{*\frac{1}{2}} = q^{-N\lambda'/n}, \qquad (4.1)$$

  *where $\lambda, \lambda' \in \mathbb{Z}\setminus\{0\}$ and $\lambda + \lambda' = 1$.*

- *for $|n| = 1, |m| > 1$:*

$$c = Nn\big(1 - \lambda(m+n)\big), \quad -p^{\frac{1}{2}} = q^{-N\lambda}, \quad -p^{*\frac{1}{2}} = q^{-Nn(1-\lambda m)}, \qquad (4.2)$$

  *where $\lambda \in \mathbb{Z}/2$ or $\lambda \in \mathbb{Z}/u$, $u$ being any divisor of $m$ or $m+n$.*

- *for $|m| = 1, |n| > 1$:*

$$c = Nm\big(\lambda'(n+m) - 1\big), \quad -p^{\frac{1}{2}} = q^{-Nm(1-\lambda'n)}, \quad -p^{*\frac{1}{2}} = q^{-N\lambda'}, \qquad (4.3)$$

  *where $\lambda' \in \mathbb{Z}/2$ or $\lambda' \in \mathbb{Z}/u'$, $u'$ being any divisor of $n$ or $n+m$.*

- *for $m = n = \pm 1$, formulas (4.2)–(4.3) also hold with $\lambda, \lambda' \in \mathbb{Z}/2$ and $\lambda + \lambda' = 1$.*

- *for $m + n = 0$ with $n > 0$ and odd:*

$$c = \frac{N}{n}, \quad -p^{\frac{1}{2}} = q^{-\frac{n-1}{2n}N}, \quad -p^{*\frac{1}{2}} = q^{-\frac{n+1}{2n}N}. \qquad (4.4)$$

## 4.2 Poisson structures on abelian quadratic subalgebras

Once the abelian subalgebras in $\mathcal{A}_{q,p}(\widehat{gl}(N)_c)$ have been characterized, one can define Poisson structures on them. Since the abelianity conditions take all the form $p = q^{\alpha N \ell}$ where $\alpha$ depends on the considered case and $\ell \in \mathbb{Z}$, we perform the expansion around a given abelianity condition by setting now $p^{1-\epsilon} = q^{\alpha N \ell}$ with $\epsilon \ll 1$ and define the corresponding Poisson structure by

$$\big\{t(z), t(w)\big\}_\ell = \lim_{\epsilon \to 0} \frac{1}{\epsilon}\big(t(z)t(w) - t(w)t(z)\big). \qquad (4.5)$$

It follows from (3.7) and the results of [16], that the Poisson structure is generically given by

$$\big\{t^{(k)}(z), t^{(k')}(w)\big\}_\ell = f_\ell^{(k,k')}(z/w)\, t^{(k)}(z)t^{(k')}(w), \qquad (4.6)$$

where

$$f_\ell^{(k,k')}(x) = \sum_{i=(1-k)/2}^{(k-1)/2} \sum_{j=(1-k')/2}^{(k'-1)/2} f_\ell(q^{i-j}x). \qquad (4.7)$$

The function $f_\ell(x)$ is given by expressions depending on the different cases of abelianity conditions of proposition 4.1. Their explicit form can be found in Theorem 4.1 of [16] (we don't recall them here for sake of brevity).

 This result allows us to consider the quantum exchange algebras (3.7) as natural quantizations (one for each surface $\mathscr{S}_{m,n}$) of these classical $q$-deformed $\mathcal{W}_N$ algebras. These are seen to include the particular classical Poisson algebra obtained by restricting the original $q$-$\mathcal{W}_N$ algebra in [1] to its sole non local elliptic exchange terms, i.e. by excluding the delta-terms. A possible explanation for this truncation arises from the fact that the quantum exchange algebra (3.7), as already discussed in section 7 of [2], is in fact a non factorized description of elliptic $\mathcal{W}_N$ algebra structures. The exact description in terms of formal series expansion in powers of the spectral parameters $z$ and $w$ requires a Riemann–Hilbert factorization of the exchange function $\mathcal{Y}(z/w)$. As explained in e.g. [2] such a Riemann–Hilbert factorization is in fact not unique, and the choice of a particular Riemann–Hilbert factorization depends upon the explicit realization of the generators $t(z)$ and the associated normal ordering prescriptions. This choice in turn implies the occurrence of extra terms resummed formally as $\delta(z/w)$-form series. Such terms may go through the classical limit and arise therefore also in the classical Poisson structures, as seen in e.g. [1]. We shall expand on this issue in the conclusion.

# 5 Critical level $c = -N$

When $m$ and $n$ take the specific values $m = 1$, $n = -1$, the surface condition leads to $c = -N$ (called the critical level) without further condition on $q$ and $p$.

## 5.1 Poisson structure at critical level

**Proposition 5.1** *At the critical level $c = -N$, the generators $t_{1,-1}^{(k)}(z)$, $k = 1, ..., N-1$, lie in the center of $\mathcal{A}_{q,p}(\widehat{gl}(N)_c)$:*

$$\left[ t_{1,-1}^{(k)}(z), L(w) \right]_{cr} = 0 \qquad k = 1, ..., N-1. \tag{5.1}$$

*Proof:* Direct consequence of relation (3.3), for $\mathcal{F}_{-1}(x) = \mathcal{U}((-p^{\frac{1}{2}})^{-1}x)$, $\mathcal{F}_{-1}^{*}(x) = \mathcal{U}((-p^{*\frac{1}{2}})^{-1}x)$ and $\mathcal{U}$ is $q^N$ periodic. Since the surface condition does not involve $p$ nor $q$, relation (5.1) is valid for the whole $\mathcal{A}_{q,p}(\widehat{gl}(N)_c)$ algebra. ∎

A natural Poisson structure can now be defined on the center. The computation requires some care, since one has to evaluate the exchange between $t_{1,-1}^{(k)}(z)$ and $t_{1,-1}^{(k')}(w)$ around $c = -N$, without assuming any particular relation between the parameters (recall that $q$ and $p$ remain generic). Setting $t_{1,-1}^{(k)}(z) = \text{tr}_{1...k}\left(Q_{1...k}(z)A_k^{(N)}\right) = \text{tr}_{1...k}\mathcal{T}^{(k)}(z)$, a direct calculation shows that

$$t_{1,-1}^{(k)}(z)\, t_{1,-1}^{(k')}(w) = \mathcal{Y}^{(k,k')}(z/w)\, \text{tr}_{1...k,1'...k'}\left(\mathscr{R}(z/w)^{-1}A_{k'}^{(N)}\right.$$
$$\left. \mathscr{R}(q^{-c-N}z/w)A_k^{(N)}\mathscr{R}(z/w)^{-1}Q_{1'...k'}(w)\mathscr{R}(q^c z/w)Q_{1...k}(z)\right), \tag{5.2}$$

where

$$\mathcal{Y}^{(k,k')}(x) = \prod_{i=(1-k)/2}^{(k-1)/2} \prod_{j=(1-k')/2}^{(k'-1)/2} \frac{\mathcal{U}(q^{i-j}x)}{\mathcal{U}(q^{i-j-c}x)} \tag{5.3}$$

and the matrix $\mathscr{R}$ is given by $\mathscr{R}(x) = \prod_{j=1'}^{\overrightarrow{k'}}\prod_{i=1}^{\overleftarrow{k}} \widehat{R}_{ij}(q^{i-j-(k-k')/2}x)$.

Using repeatedly the relations (6.10) and (6.14), explicitly proved in the next section, one checks that

$$\mathscr{R}(x)A_k^{(N)} = A_k^{(N)}\mathscr{R}(x)A_k^{(N)}, \qquad \mathscr{R}^{-1}(x)A_k^{(N)} = A_k^{(N)}\mathscr{R}^{-1}(x)A_k^{(N)}, \tag{5.4}$$

$$\mathscr{R}(x)A_{k'}^{(N)} = A_{k'}^{(N)}\mathscr{R}(x)A_{k'}^{(N)}, \qquad \mathscr{R}^{-1}(x)A_{k'}^{(N)} = A_{k'}^{(N)}\mathscr{R}^{-1}(x)A_{k'}^{(N)}. \tag{5.5}$$

The fused matrix $\mathscr{R}(x)$ also satisfies the crossing-unitarity relation, $T$ denoting the transposition in the first spaces (i.e. $T = t_1 \ldots t_k$):

$$\left(\mathscr{R}(x)^T\right)^{-1} = \left(\mathscr{R}(q^N x)^{-1}\right)^T. \tag{5.6}$$

Consider the trace in spaces $1', ..., k'$ in the LHS of (5.2). Applying successively the relations (5.5) and (6.15), a copy of the antisymmetrizer $A_{k'}^{(N)}$ is reproduced all the way to the left; the leftmost term is moved to the right by cyclicity of the trace, and then reproduced again to the left until it re-absorbs the antisymmetrizer $A_{k'}^{(N)}$ in the middle of the expression. Every $A_{k'}^{(N)}$ term from left to right is then indeed eliminated until only the rightmost term remains:

$$\text{tr}_{1'...k'}\left(\mathscr{R}(z/w)^{-1}A_{k'}^{(N)}\mathscr{R}(q^{-c-N}z/w)A_k^{(N)}\mathscr{R}(z/w)^{-1}Q_{1'...k'}(w)\mathscr{R}(q^c z/w)Q_{1...k}(z)\right)$$
$$= \text{tr}_{1'...k'}\left(\mathscr{R}(z/w)^{-1}\mathscr{R}(q^{-c-N}z/w)A_k^{(N)}\mathscr{R}(z/w)^{-1}Q_{1'...k'}(w)\mathscr{R}(q^c z/w)A_{k'}^{(N)}Q_{1...k}(z)\right). \tag{5.7}$$

Exchanging the roles of the traces over the spaces $1'\ldots k'$ and $1\ldots k$, a similar strategy is used to move the antisymmetrizer $A_k^{(N)}$ to the rightmost thanks to (5.4):

$$\text{tr}_{1\ldots k}\left(\mathscr{R}(z/w)^{-1}\,\mathscr{R}(q^{-c-N}z/w)A_k^{(N)}\mathscr{R}(z/w)^{-1}Q_{1'\ldots k'}(w)\mathscr{R}(q^c z/w)A_{k'}^{(N)}Q_{1\ldots k}(z)\right)$$
$$=\text{tr}_{1\ldots k}\left(\mathscr{R}(z/w)^{-1}\,\mathscr{R}(q^{-c-N}z/w)\mathscr{R}(z/w)^{-1}Q_{1'\ldots k'}(w)\mathscr{R}(q^c z/w)A_{k'}^{(N)}Q_{1\ldots k}(z)A_k^{(N)}\right). \quad (5.8)$$

Now, from the property

$$\text{tr}_{1'\ldots k'}\left(\mathcal{O}_{1\ldots k,1'\ldots k'}\mathcal{M}_{1\ldots k,1'\ldots k'}\right)=\text{tr}_{1'\ldots k'}\left(\left(\mathcal{M}_{1\ldots k,1'\ldots k'}\right)^T\left(\mathcal{O}_{1\ldots k,1'\ldots k'}\right)^T\right)^T,$$

where $\mathcal{O}$ is operator-like and $\mathcal{M}$ is a numerical matrix, one has

$$\text{tr}_{1'\ldots k'}\left(\mathscr{R}(z/w)^{-1}\,\mathscr{R}(q^{-c-N}z/w)\mathscr{R}(z/w)^{-1}Q_{1'\ldots k'}(w)\mathscr{R}(q^c z/w)A_{k'}^{(N)}\right)$$
$$=\text{tr}_{1'\ldots k'}\left(\left(\mathscr{R}(q^c z/w)A_{k'}^{(N)}\right)^T\left(\mathscr{R}(z/w)^{-1}\,\mathscr{R}(q^{-c-N}z/w)\mathscr{R}(z/w)^{-1}Q_{1'\ldots k'}(w)\right)^T\right)^T$$
$$=\text{tr}_{1'\ldots k'}\left(\left(\left(\mathscr{R}(q^c z/w)\right)^T\left(\mathscr{R}(z/w)^{-1}\,\mathscr{R}(q^{-c-N}z/w)\mathscr{R}(z/w)^{-1}\right)^T\right)^T Q_{1'\ldots k'}(w)A_{k'}^{(N)}\right), \quad (5.9)$$

where we used again the above trick to bring the antisymmetrizer $A_{k'}^{(N)}$ to the right in the last line, since the relation (5.5) is also valid for the matrix $\mathscr{R}(x)^T$.

Taking now into account the trace over the spaces $1\ldots k$, one obtains

$$t_{1,-1}^{(k)}(z)\,t_{1,-1}^{(k')}(w)=\mathcal{Y}^{(k,k')}(z/w)\,\text{tr}_{1\ldots k,1'\ldots k'}\left(\mathcal{M}(z/w)\,\mathcal{T}^{(k')}(w)\,\mathcal{T}^{(k)}(z)\right), \quad (5.10)$$

with

$$\mathcal{M}(x)=\left(\mathscr{R}(q^c x)^T\left(\mathscr{R}(x)^{-1}\mathscr{R}(q^{-c-N}x)\mathscr{R}(x)^{-1}\right)^T\right)^T. \quad (5.11)$$

When $c=-N$, one gets immediately $\mathcal{Y}^{(k,k')}(x)|_{cr}=1$ and $\mathcal{M}(x)|_{cr}=\mathbb{I}$. Hence one directly recovers the commutativity of the generators $t_{1,-1}^{(k)}(z)$ and $t_{1,-1}^{(1)}(w)$ at the critical level (according to corollary 5.1). The Poisson bracket is then defined by

$$\left\{t_{1,-1}^{(k)}(z),\,t_{1,-1}^{(k')}(w)\right\}_{cr}=\text{tr}_{1\ldots k,1'\ldots k'}\left(\frac{d}{dc}\left(\mathcal{Y}^{(k,k')}(z/w)\mathcal{M}(z/w)\right)\bigg|_{cr}\mathcal{T}^{(k')}(w)\mathcal{T}^{(k)}(z)\right). \quad (5.12)$$

One has

$$\frac{d}{dc}\left(\mathcal{Y}^{(k,k')}(x)\mathcal{M}(x)\right)\bigg|_{cr}=\frac{d}{dc}\mathcal{Y}^{(k,k')}(x)\bigg|_{cr}\mathbb{I}+\frac{d}{dc}\mathcal{M}(x)\bigg|_{cr} \quad (5.13)$$

and

$$\frac{d}{dc}\mathcal{M}^T(x)\bigg|_{cr}=\frac{d}{dc}\mathscr{R}(q^c x)^T\left(\mathscr{R}(x)^{-1}\right)^T+\mathscr{R}(q^{-N}x)^T\left(\mathscr{R}(x)^{-1}\frac{d}{dc}\mathscr{R}(q^{-c-N}x)\mathscr{R}(x)^{-1}\right)^T\bigg|_{cr}$$
$$=\frac{d}{dc}\mathscr{R}(q^c x)^T\left(\mathscr{R}(q^c x)^T\right)^{-1}-\mathscr{R}(q^c x)^T\frac{d}{dc}\left(\mathscr{R}^{-1}(q^{-c-N}x)\right)^T\bigg|_{cr}. \quad (5.14)$$

Since $\dfrac{d}{dc}\mathscr{R}^{-1}(q^{-c-N}x)\bigg|_{cr}=-\dfrac{d}{dc}\mathscr{R}^{-1}(q^{c+N}x)\bigg|_{cr}$ and using the crossing-unitarity property for $\mathscr{R}$, one gets

$$\frac{d}{dc}\mathcal{M}^T(x)\bigg|_{cr}=\frac{d}{dc}\mathscr{R}(q^c x)^T\left(\mathscr{R}(q^c x)^T\right)^{-1}+\mathscr{R}(q^c x)^T\frac{d}{dc}\left(\mathscr{R}(q^c x)^T\right)^{-1}\bigg|_{cr}. \quad (5.15)$$

It follows that $\left.\dfrac{d}{dc}\mathcal{M}(x)\right|_{cr} = 0$. This guarantees that the Poisson bracket (5.12) indeed closes on $t_{1,-1}^{(k)}(z)$ and $t_{1,-1}^{(k')}(w)$, a property which was not obvious:

$$\left\{t_{1,-1}^{(k)}(z), t_{1,-1}^{(k')}(w)\right\}_{cr} = \left.\frac{d}{dc}\mathcal{Y}^{(k,k')}(z/w)\right|_{cr} t_{1,-1}^{(k')}(w)\, t_{1,-1}^{(k)}(z). \tag{5.16}$$

Finally the derivative $\left.\dfrac{d}{dc}\mathcal{Y}^{(k,k')}(x)\right|_{cr}$ is explicitly computed. Since $\mathcal{Y}^{(k,k')}(x)|_{cr} = 1$, using the definition of $\mathcal{U}(x)$ in terms of $\tau_N(x)$ and its $q^N$-periodicity, one obtains

$$\left.\frac{d}{dc}\mathcal{Y}^{(k,k')}(x)\right|_{cr} = -\ln q \sum_{i=(1-k)/2}^{(k-1)/2} \sum_{j=(1-k')/2}^{(k'-1)/2} \left.\left(y\frac{d}{dy}\ln\tau_N(q^{\frac{1}{2}}y) - y^{-1}\frac{d}{dy^{-1}}\ln\tau_N(q^{\frac{1}{2}}y^{-1})\right)\right|_{y=xq^{i-j}}. \tag{5.17}$$

Therefore, after evaluating the logarithmic derivatives of the function $\tau_N$, see e.g. [9], and denoting $f_{cr}^{(k,k')}(x) = \left.\dfrac{d}{dc}\mathcal{Y}^{(k,k')}(x)\right|_{cr}$, one gets the following result:

**Proposition 5.2** *At the critical level $c = -N$, the elements $t_{1,-1}^{(k)}(z)$ form a closed algebra under the Poisson bracket on the center of $\mathcal{A}_{q,p}(\widehat{gl}(N)_c)$ given by*

$$\left\{t_{1,-1}^{(k)}(z), t_{1,-1}^{(k')}(w)\right\}_{cr} = f_{cr}^{(k,k')}(z/w)\, t_{1,-1}^{(k')}(w)\, t_{1,-1}^{(k)}(z), \tag{5.18}$$

*where*

$$f_{cr}^{(k,k')}(x) = -2\ln q \sum_{i=(1-k)/2}^{(k-1)/2} \sum_{j=(1-k')/2}^{(k'-1)/2} \left(2I(q^{i-j}x) - I(q^{i-j+1}x) - I(q^{i-j-1}x)\right), \tag{5.19}$$

*with*

$$I(x) = \sum_{\ell \geq 0} \frac{x^2 q^{2N\ell}}{1 - x^2 q^{2N\ell}} - \frac{1}{2}\frac{x^2}{1-x^2} - (x \leftrightarrow x^{-1}). \tag{5.20}$$

Using the results of the Appendix B of [9], one can recast the formula (5.19) as follows:

$$f_{cr}^{(k,k')}(x) = -2(q-q^{-1})\ln q \sum_{r\in\mathbb{Z}} \frac{[(N-\max(k,k'))r]_q\,[\min(k,k')r]_q}{[Nr]_q}\, x^{2r}, \tag{5.21}$$

where $[n]_q = \dfrac{q^n - q^{-n}}{q - q^{-1}}$ denotes the $q$-number.

The expression (5.21) of the structure function $f_{cr}^{(k,k')}(x)$ allows one to recover the 'leading term' (i.e. up to delta terms) of the Poisson structure found in [1]. Recall that the latter is obtained in the framework of the quantum affine algebra $\mathcal{U}_q(\widehat{gl}_N)_{cr}$ at the critical level, which exhibits an extended center. The Wakimoto realization of the algebra provides a $q$-deformation of the Miura transformation. When applied on the extended center of $\mathcal{U}_q(\widehat{gl}_N)_{cr}$, one gets the desired Poisson structure.

It may be possible to reconstruct the delta-type terms directly in the Poisson algebra, starting from the initial structure (5.19), without explicitly starting from the quantum "Riemann–Hilbert splitted" algebra discussed in section 4. Indeed in the case of DVA when $N = 2$, it can be shown that the generator defined by $s_1(z) = t_{1,-1}^{(1)}(q^{-1/2}z) + t_{1,-1}^{(1)}(q^{1/2}z)^{-1}$ has the following Poisson structure:

$$\left\{s_1(z), s_1(w)\right\}_{cr} = f_{cr}^{(1,1)}(z/w)\, s_1(w)s_1(z) + \delta(qz/w) - \delta(qw/z), \tag{5.22}$$

hence one recovers the centrally extended DVA constructed by Frenkel and Reshetikhin [1]. The question of generalizing such generators $s_k(z)$ ($k = 1, ..., N-1$) in order to recover the $q$-$\mathcal{W}_N$ algebra included the delta terms of [2] remains however open.

## 5.2 Degeneration to $\mathcal{U}_q(\widehat{gl}_N)$ at critical level

**Proposition 5.3** *In the limit $p \to 0$, the construction described in theorem 3.1 reduces to the construction done in [14] for the algebra $\mathcal{U}_q(\widehat{gl}(N))$. The limit $p \to 0$ of the Poisson structure (5.18) is the Poisson structure of proposition 5.2.*

*Proof:* We introduce objects corresponding to the monodromy matrices of $\mathcal{U}_q(\widehat{gl}_N)$ in the limit $p \to 0$:

$$L^+(z^2) = V(q^{c/2}z)^{-1} F^{-1} L(q^{c/2}z) F^{-1} V(q^{c/2}z) \tag{5.23}$$

$$L^-(z^2) = V(z)^{-1} F^{-1} (g^{\frac{1}{2}} h g^{\frac{1}{2}}) L(-p^{\frac{1}{2}}z) (g^{\frac{1}{2}} h g^{\frac{1}{2}})^{-1} F^{-1} V(z), \tag{5.24}$$

where we have introduced the twist between the presentations of $\mathcal{U}_q(\widehat{gl}_N)$ corresponding to principal gradation and the non-elliptic limit,

$$F = \sum_{j=1}^{N} \mathfrak{f}_j e_{jj} \quad \text{with} \quad \mathfrak{f}_j = q^{-\sum_{i=1}^{N} \alpha_{ji} h_i}, \quad \alpha_{ij} = \begin{cases} \frac{1}{2} + \frac{i-j}{N} & \text{if } i < j \\ 0 & \text{if } i = j \\ -\alpha_{ji} & \text{if } i > j \end{cases}$$

and the gauge transformation that relates the principal and the homogeneous gradations

$$V(z) = \sum_{j=1}^{N} z^{\frac{N+1-2j}{N}} e_{jj}$$

(for details see [15]). It allows us to rewrite $t^{(k)}_{1,-1}(z)$ as

$$t^{(k)}_{1,-1}(z) = \operatorname{tr}_{1\ldots k} \Big[ \prod_{i=1}^{\overleftarrow{k}} \Big( V_i\big(-p^{\frac{1}{2}} q^{-N/2} z_i\big) F_i L_i^+\big(p q^{-N} z_i^2\big) F_i \Big) \times$$

$$\times \prod_{i=1}^{\overrightarrow{k}} \Big( F_i^{-1} L_i^-(p z_i^2)^{-1} F_i^{-1} V_i\big(-p^{\frac{1}{2}} q^{N/2} z_i\big)^{-1} \Big) A_k^{(N)} \Big], \tag{5.25}$$

where we have simplified the $V_i$'s in between the two products. We need the relations

$$\mathfrak{f}_i L_{jl}^+(z) = L_{jl}^+(z) \mathfrak{f}_i q^{\alpha_{ij} - \alpha_{il}} \quad \text{and} \quad \mathfrak{f}_i L_{jl}^-(z)^{-1} = L_{jl}^-(z)^{-1} \mathfrak{f}_i q^{\alpha_{ij} - \alpha_{il}}$$

together with

$$\sum_{1 \le a < b \le k} \alpha_{j_{\sigma(a)} j_{\sigma(b)}} + \sum_{a=1}^{k} \frac{2a}{N} (j_{\sigma(a)} - j_a) = -\ell(\sigma) + \sum_{1 \le a < b \le k} \alpha_{j_a j_b}, \tag{5.26}$$

where $\ell(\sigma)$ is the length of $\sigma$. Relation (5.26) is valid for any permutation $\sigma \in S_k$ and any set of indices $\{j_1, \ldots, j_k\} \subset \{1, \ldots, N\}$. Then, from these relations, and using cyclicity one can rewrite (5.25) as

$$t^{(k)}_{1,-1}(z) = \operatorname{tr}_{1\ldots k} \Big[ \prod_{i=1}^{\overleftarrow{k}} L_i^+\big(p q^{-N} z_i^2\big) \prod_{i=1}^{\overrightarrow{k}} L_i^-(p z_i^2)^{-1} \prod_{i=1}^{k} D_i A_k^q \Big], \tag{5.27}$$

where $D = \operatorname{diag}(q^{1-N}, q^{3-N}, \ldots, q^{N-3}, q^{N-1})$ and $A_k^q$ is the quantum antisymmetrizer. One recognizes the expression given in [14], with the change $q \to q^{-1}$ (due to a different definition of the $R$-matrix in $\mathcal{U}_q(\widehat{gl}_N)$).

The $p \to 0$ degeneration of the $t^{(k)}(z)$ generators at the critical level can be applied *mutatis mutandis* to the evaluation of the exchange relation between these generators when $p \to 0$. The result is that proposition 5.2 also holds for the $t^{(k)}(z)$ generators in the $\mathcal{U}_q(\widehat{gl}_N)_{cr}$ case. Note *en passant* that the structure function $f_{cr}^{(k,k')}(x)$ is $p$-independent. ∎

# 6 Proof of Theorem 3.1

We start by considering the following operator

$$t^{(k)}(z) = \mathrm{tr}_{1\ldots k}\left(\mathbb{M}\,\mathcal{L}_{1\ldots k}(z;\bar\alpha)\,\tilde{\mathbb{M}}\,\mathcal{L}_{1\ldots k}(z;\bar\beta)^{-1}A_k^{(N)}\right) := \mathrm{tr}_{1\ldots k}\left(Q_{1\ldots k}(z)A_k^{(N)}\right), \tag{6.1}$$

where $A_k^{(N)}$ is the antisymmetrizer (acting in spaces $1,\ldots,k$) and the matrices $\mathbb{M}$ and $\tilde{\mathbb{M}}$ are defined in theorem 3.1. For any set of parameters $\bar\alpha = \{\alpha_1,\ldots,\alpha_k\}$, we introduce the notation:

$$\mathcal{L}(z;\bar\alpha) = \prod_{i=1}^{\overleftarrow{k}} L_i(\alpha_i z). \tag{6.2}$$

By successive use of the *RLL* relation (2.3), one gets

$$t^{(k)}(z)L_0(w)^{-1} = \mathrm{tr}_{1\ldots k}\left(\mathbb{M}\mathcal{L}(z;\bar\alpha)\tilde{\mathbb{M}}\,\mathcal{R}^*_{1\ldots k,0}(\tfrac{z}{w};\bar\beta)\,L_0(w)^{-1}\mathcal{L}(z;\bar\beta)^{-1}\,\mathcal{R}^{-1}_{1\ldots k,0}(\tfrac{z}{w};\bar\beta)A_k^{(N)}\right),$$

where we have introduced

$$\mathcal{R}^*_{1\ldots k,0}(x;\bar\beta) = \prod_{i=1}^{\overleftarrow{k}} \widehat{R}^*_{i0}(\beta_i x) \quad\text{and}\quad \mathcal{R}^{-1}_{1\ldots k,0}(x;\bar\beta) = \prod_{i=1}^{\overrightarrow{k}} \widehat{R}^{-1}_{i0}(\beta_i x).$$

Due to the quasi-periodicity property of the *R*-matrix (see [16]), we have

$$\tilde{M}_i\widehat{R}^*_{i0}(x) = \mathcal{F}^*_n(x)\widehat{R}^*_{i0}(\gamma^* x)\tilde{M}_i \quad\text{with}\quad \gamma^* = (-p^{*\frac{1}{2}})^n, \tag{6.3}$$

$$M_i\widehat{R}_{i0}(x) = \mathcal{F}_m(x)\widehat{R}_{i0}(\gamma x)M_i \quad\text{with}\quad \gamma = (-p^{\frac{1}{2}})^m, \tag{6.4}$$

where the functions $\mathcal{F}^*_n(x)$ and $\mathcal{F}_m(x)$ are given in (3.5). Then, one has

$$t^{(k)}(z)L_0(w)^{-1} = \prod_{i=1}^{k}\mathcal{F}^*_n(\beta_i z/w)\,\mathrm{tr}_{1\ldots k}\left(\mathbb{M}\mathcal{L}(z;\bar\alpha)\mathcal{R}^*_{1\ldots k,0}(\tfrac{z}{w};\gamma^*\bar\beta)\,L_0(w)^{-1}\right.$$

$$\left. \tilde{\mathbb{M}}\,\mathcal{L}(z;\bar\beta)^{-1}\,\mathcal{R}^{-1}_{1\ldots k,0}(\tfrac{z}{w};\bar\beta)A_k^{(N)}\right)$$

$$= \prod_{i=1}^{k}\mathcal{F}^*_n(\beta_i z/w)\,L_0(w)^{-1}\,\mathrm{tr}_{1\ldots k}\left(\mathbb{M}\mathcal{R}_{1\ldots k,0}(\tfrac{z}{w};\gamma^*\bar\beta)\,\mathcal{L}(z;\alpha)\right.$$

$$\left. \tilde{\mathbb{M}}\,\mathcal{L}(z;\bar\beta)^{-1}\,\mathcal{R}^{-1}_{1\ldots k,0}(\tfrac{z}{w};\bar\beta)A_k^{(N)}\right). \tag{6.5}$$

In the second equality, $\gamma^*\bar\beta = \{\gamma^*\beta_1,\ldots,\gamma^*\beta_k\}$ and the parameters are supposed to fulfill the condition $\alpha_i = \gamma^*\beta_i$ so that we can use the RLL relation (2.3). Using relation (6.4) we obtains

$$t^{(k)}(z)L_0(w)^{-1} = \varphi(z/w)L_0(w)^{-1}\,\mathrm{tr}_{1\ldots k}\left(\mathcal{R}_{1\ldots k,0}(\tfrac{z}{w};\gamma\gamma^*\bar\beta)Q_{1\ldots k}(z)\mathcal{R}^{-1}_{1\ldots k,0}(\tfrac{z}{w};\bar\beta))A_k^{(N)}\right), \tag{6.6}$$

the proportionality coefficient being given by

$$\varphi(x) = \prod_{i=1}^{k}\mathcal{F}^*_n(\beta_i x)\mathcal{F}_m(\gamma^*\beta_i x) = \prod_{i=1}^{k}\frac{\mathcal{F}^*_n(\beta_i x)}{\mathcal{F}_{-m}(\beta_i x)}. \tag{6.7}$$

The following lemma provides a sufficient condition for the RHS of (6.6) to build back the $t^{(k)}(z)$ operator.

**Lemma 6.1** *When the relations $\gamma\gamma^* = q^{-N}$ and $\beta_i = q\beta_{i-1}$ are fulfilled, we have*

$$\text{tr}_{1\ldots k}\left(\mathcal{R}_{1\ldots k,0}(x;\gamma\gamma^*\bar\beta)Q_{1\ldots k}(z)\mathcal{R}^{-1}_{1\ldots k,0}(x;\bar\beta)A_k^{(N)}\right) = \text{tr}_{1\ldots k}\left(Q_{1\ldots k}(z)A_k^{(N)}\right), \quad \forall x,z. \quad (6.8)$$

**Proof:** Consider the antisymmetrizer $A_k^{(N)}$ on $(\mathbb{C}^N)^{\otimes k}$. Since $\ker\widehat{R}_{12}(q) = \text{im}A_2^{(N)}$ (see [15, 20]), one has $\bigcap_{i=1}^{k-1}\ker\widehat{R}_{i,i+1}(q) = \text{im}A_k^{(N)}$. By action of $\widehat{R}_{i,i+1}(q)$ ($i = 1,\ldots,k-1$) on the product $L_1(z)\ldots L_k(zq^{1-k})$, one deduces that

$$L_1(z)\ldots L_k(zq^{1-k})A_k^{(N)} = A_k^{(N)}L_1(z)\ldots L_k(zq^{1-k})A_k^{(N)}. \quad (6.9)$$

The $R$-matrices $\widehat{R}_{i0}(z)$ and $(\widehat{R}^{-1}_{i0})^{t_0}(z)$ being representations of the $L$-matrices $L_i(z)$, one gets

$$\widehat{R}_{10}(z)\ldots\widehat{R}_{k0}(zq^{1-k})A_k^{(N)} = A_k^{(N)}\widehat{R}_{10}(z)\ldots\widehat{R}_{k0}(zq^{1-k})A_k^{(N)}, \quad (6.10)$$

$$(\widehat{R}^{-1}_{10})^{t_0}(z)\ldots(\widehat{R}^{-1}_{k0})^{t_0}(zq^{1-k})A_k^{(N)} = A_k^{(N)}(\widehat{R}^{-1}_{10})^{t_0}(z)\ldots(\widehat{R}^{-1}_{k0})^{t_0}(zq^{1-k})A_k^{(N)}. \quad (6.11)$$

Moreover, starting with the relations (2.3) written as

$$\widehat{R}^*_{12}(z/w)L'_1(z)L'_2(w) = L'_2(w)L'_1(z)\widehat{R}_{12}(z/w), \quad (6.12)$$

where $L'(z) = L(z^{-1})^{-1}$, and using the ker property on $\widehat{R}^*_{12}$, one now gets a similar property on the Lax matrices $L(z)^{-1}$:

$$L_1(z)^{-1}\ldots L_k(zq^{k-1})^{-1}A_k^{(N)} = A_k^{(N)}L_1(z)^{-1}\ldots L_k(zq^{k-1})^{-1}A_k^{(N)}. \quad (6.13)$$

When expressed in terms of the $R$-matrix representation, this relation implies

$$R^{-1}_{10}(z)\ldots R^{-1}_{k0}(zq^{k-1})A_k^{(N)} = A_k^{(N)}R^{-1}_{10}(z)\ldots R^{-1}_{k0}(zq^{k-1})A_k^{(N)}. \quad (6.14)$$

Since one has $M_1\ldots M_kA_k^{(N)} = A_k^{(N)}M_1\ldots M_k$ (with a similar relation for the matrices $\tilde{M}_i$), one then deduces, with the help of (6.9) and (6.13), the following relation, provided that $\beta_i = q\beta_{i-1}$:

$$Q_{1\ldots k}(z)A_k^{(N)} = \prod_{i=1}^{\overleftarrow{k}}M_i\,L_i(\alpha_i z)\cdot\prod_{i=1}^{\overrightarrow{k}}\tilde{M}_i\,L_i(\beta_i z)^{-1}A_k^{(N)} = A_k^{(N)}Q_{1\ldots k}(z)A_k^{(N)}. \quad (6.15)$$

Now, one has

$$\text{tr}_{1\ldots k}\left(\mathcal{R}_{1\ldots k,0}(x;\gamma\gamma^*\bar\beta)Q_{1\ldots k}(z)\mathcal{R}^{-1}_{1\ldots k,0}(x;\bar\beta)A_k^{(N)}\right)$$

$$= \text{tr}_{1\ldots k}\left(Q_{1\ldots k}(z)\mathcal{R}^{-1}_{1\ldots k,0}(x;\bar\beta)^{t_0}A_k^{(N)}\mathcal{R}_{1\ldots k,0}(x;\gamma\gamma^*\bar\beta)^{t_0}\right)^{t_0} \quad (6.16)$$

$$= \text{tr}_{1\ldots k}\left(Q_{1\ldots k}(z)\prod_{i=1}^{\overleftarrow{k}}\left(\widehat{R}^{-1}_{i0}(\beta_i x)\right)^{t_0}A_k^{(N)}\prod_{i=1}^{\overrightarrow{k}}\widehat{R}^{t_0}_{i0}(\gamma\gamma^*\beta_i x)\right)^{t_0}$$

$$= \text{tr}_{1\ldots k}\left(Q_{1\ldots k}(z)A_k^{(N)}\prod_{i=1}^{\overrightarrow{k}}\left(\widehat{R}^{-1}_{i0}(\beta_{k+1-i}x)\right)^{t_0}A_k^{(N)}\prod_{i=1}^{\overrightarrow{k}}\widehat{R}^{t_0}_{i0}(\gamma\gamma^*\beta_i x)\right)^{t_0} \quad (6.17)$$

$$= \text{tr}_{1\ldots k}\left(Q_{1\ldots k}(z)A_k^{(N)}\prod_{i=1}^{\overrightarrow{k}}\left(\widehat{R}^{-1}_{i0}(\beta_{k+1-i}x)\right)^{t_0}A_k^{(N)}\prod_{i=1}^{\overrightarrow{k}}\widehat{R}^{t_0}_{i0}(\gamma\gamma^*\beta_i x)A_k^{(N)}\right)^{t_0} \quad (6.18)$$

$$= \text{tr}_{1\ldots k}\left(Q_{1\ldots k}(z)A_k^{(N)}\prod_{i=1}^{\overrightarrow{k}}\left(\widehat{R}^{-1}_{i0}(\beta_{k+1-i}x)\right)^{t_0}A_k^{(N)}\prod_{i=1}^{\overleftarrow{k}}\widehat{R}^{t_0}_{i0}(\gamma\gamma^*\beta_{k+1-i}x)A_k^{(N)}\right)^{t_0}. \quad (6.19)$$

We used in (6.16) the relation $\mathrm{tr}_{1...k}\left(A_{1...k0}B_{1...k0}\right) = \mathrm{tr}_{1...k}\left(\left(B_{1...k0}\right)^{t_0}\left(A_{1...k0}\right)^{t_0}\right)^{t_0}$, in (6.17) the relation (6.11) together with a reordering of the matrices, and in (6.18) the relation (6.15) and the cyclicity of the trace.

Applying the transposition $t_0$ to the relation (6.10), one gets

$$A_k^{(N)}\prod_{i=1}^{\overleftarrow{k}}\widehat{R}_{i0}^{t_0}(\gamma\gamma^*\beta_{k+1-i}x)A_k^{(N)} = \prod_{i=1}^{\overleftarrow{k}}\widehat{R}_{i0}^{t_0}(\gamma\gamma^*\beta_{k+1-i}x)A_k^{(N)}, \tag{6.20}$$

and therefore

$$\mathrm{tr}_{1...k}\left(\mathcal{R}_{1...k,0}(x;\gamma\gamma^*\bar{\beta})Q_{1...k}(z)\mathcal{R}_{1...k,0}^{-1}(x;\bar{\beta})A_k^{(N)}\right)$$

$$= \mathrm{tr}_{1...k}\left(Q_{1...k}(z)A_k^{(N)}\prod_{i=1}^{\overrightarrow{k}}\left(\widehat{R}_{i0}^{-1}(\beta_{k+1-i}x)\right)^{t_0}\prod_{i=1}^{\overleftarrow{k}}\widehat{R}_{i0}^{t_0}(\gamma\gamma^*\beta_{k+1-i}x)A_k^{(N)}\right)^{t_0}. \tag{6.21}$$

The two products of $R$-matrices in the RHS simplify thanks to the crossing-unitarity property (B.16) provided $\gamma\gamma^* = q^{-N}$. Then the expression rewrites

$$\mathrm{tr}_{1...k}\left(Q_{1...k}(z)(A_k^{(N)})^2\right)^{t_0} = \mathrm{tr}_{1...k}\left(Q_{1...k}(z)A_k^{(N)}\right).$$

This concludes the proof of lemma 6.1.

Taking into account lemma 6.1, one gets

$$t_{m,n}^{(k)}(z)L_0(w)^{-1} = \varphi(z/w)L_0(w)^{-1}t_{m,n}^{(k)}(z), \tag{6.22}$$

where the indices $m, n$ remind that the above equation is valid on the surface $\mathscr{S}_{m,n}$ and the function $\varphi(x)$ is explicited in (6.7). Given the values of the parameters $\beta_i, \gamma_i, \gamma_i^*$, one finally obtains (3.3). ∎

# 7  Conclusion

We wish to conclude with a few remarks regarding questions and issues which have been opened by the present work.

A first remark concerns the quantum $q$-$\mathcal{W}_N$ algebra structures proposed in this paper: they define consistently an abstract algebra, whose exchange relations preserve the spin content of the $W$ generators. Obviously, the next step should be to find explicit realization(s) of these algebras, based on known structures, such as (deformed) oscillator algebras. For instance, one can build an explicit realization of the algebra by vertex-operator construction as in [2], with suitable modifications to account for the now different forms of the exchange functions. This could help in establishing possible physical interpretations of these new algebras.

Most probably, as in the $\mathcal{W}_{q,p}(A_N)$ case such construction will involve lower spin local terms (i.e. of the form $\delta(\alpha z/w)$), associated to the normal ordering of the vertex operators and formally understood as realizing distinct Riemann–Hilbert factorizations of the same elliptic exchange kernel. The question is then to determine whether these terms are only of central extension type, or whether non central, lower-spin terms also arise, yielding an *a priori* different $q$-$\mathcal{W}_N$ algebra. One should then investigate the possibility to add lower degree counter terms to the realization of the $W$ generators in order to recover the same $q$-$\mathcal{W}_N$ algebra up to central terms. The existence of such counter terms is not guaranteed at all, and the whole

construction remains an open question. One can alternatively try to propose an ansatz for such local terms and solve the associativity conditions for these assumed exchange relations. This was indeed partially achieved in the DVA case [21] but is known to entail intricate technical manipulations. Of course similar issues arise regarding the extra terms in the $q$-deformed Poisson structures.

A second issue arises, as was already the case in the DVA context [16], regarding the new "unitary" elliptic algebra introduced in (3.9). It appears to be a well-defined elliptic algebra structure, but its precise understanding as a quasi-Hopf algebra, and its eventual connection to quantum affine algebras (possibly by a twist) remains elusive. Note in this respect that at this stage one even lacks an explicit computational link[1], when $N > 2$, between the universal $R$-matrix of $\mathcal{A}_{q,p}(\widehat{gl}(N)_c)$ [12, 22] and the actual elliptic Belavin–Baxter vertex $R$-matrix in Appendix B. In fact, from the non-elliptic limit and the twist needed (when $N > 2$) to connect it to the usual presentations of $U_q(gl_N)$ [15], it seems rather unavoidable that a twist by a cocycle will be needed to connect the construction [12,22] to the actual elliptic Belavin–Baxter vertex $R$-matrix for $N > 2$.

A structural issue regarding these new $q$-$\mathcal{W}_N$ algebras would also be whether they can be embedded in some representation (possibly non-horizontal) of the Ding–Iohara–Miki toroidal algebra [23][2], as is the case for the canonical Feigin and Frenkel $q$-$\mathcal{W}_N$ algebra [24].

We finally wish to point out an interesting recent construction of elliptic $W$-algebras [25], using methods of quiver theory. The authors use vertex-algebra type constructions to yield exchange relations formulated in terms of elliptic Gamma functions. This is an extension to $A_N$ algebras of the Nieri construction [26] of elliptic Virasoro algebras, and also relates to $W_{q,t}$ algebras defined by Frenkel and Reshetikin [27]. A comparison between these and our structures would certainly be interesting.

## A Jacobi theta functions

Let $\mathbb{H} = \{z \in \mathbb{C} \,|\, \mathrm{Im}\, z > 0\}$ be the upper half-plane and $\Lambda_\tau = \{\lambda_1 \tau + \lambda_2 \,|\, \lambda_1, \lambda_2 \in \mathbb{Z}, \tau \in \mathbb{H}\}$ the lattice with basis $(1, \tau)$ in the complex plane. One denotes the congruence ring modulo $N$ by $\mathbb{Z}_N \equiv \mathbb{Z}/N\mathbb{Z}$ with basis $\{0, 1, \ldots, N-1\}$. One sets $\omega = e^{2i\pi/N}$.

One defines the Jacobi theta functions with rational characteristics $(\gamma_1, \gamma_2) \in \frac{1}{N}\mathbb{Z} \times \frac{1}{N}\mathbb{Z}$ by:

$$\vartheta\begin{bmatrix} \gamma_1 \\ \gamma_2 \end{bmatrix}(\xi, \tau) = \sum_{m \in \mathbb{Z}} \exp\left(i\pi(m + \gamma_1)^2 \tau + 2i\pi(m + \gamma_1)(\xi + \gamma_2)\right). \tag{A.1}$$

The functions $\vartheta\begin{bmatrix} \gamma_1 \\ \gamma_2 \end{bmatrix}(\xi, \tau)$ satisfy the following shift properties:

$$\vartheta\begin{bmatrix} \gamma_1 + \lambda_1 \\ \gamma_2 + \lambda_2 \end{bmatrix}(\xi, \tau) = \exp(2i\pi\gamma_1\lambda_2)\,\vartheta\begin{bmatrix} \gamma_1 \\ \gamma_2 \end{bmatrix}(\xi, \tau), \tag{A.2}$$

$$\vartheta\begin{bmatrix} \gamma_1 \\ \gamma_2 \end{bmatrix}(\xi + \lambda_1\tau + \lambda_2, \tau) = \exp\left(-i\pi\lambda_1^2\tau - 2i\pi\lambda_1\xi + 2i\pi(\gamma_1\lambda_2 - \gamma_2\lambda_1)\right)\vartheta\begin{bmatrix} \gamma_1 \\ \gamma_2 \end{bmatrix}(\xi, \tau), \tag{A.3}$$

where $(\gamma_1, \gamma_2) \in \frac{1}{N}\mathbb{Z} \times \frac{1}{N}\mathbb{Z}$ and $(\lambda_1, \lambda_2) \in \mathbb{Z} \times \mathbb{Z}$.
Moreover, for arbitrary $(\lambda_1, \lambda_2)$ (not necessarily integers), one has the following shift ex-

---

[1]We wish to thank H. Konno for comments on this issue.
[2]We thank the referee for mentioning this possibility.

change:

$$\vartheta\begin{bmatrix}\gamma_1\\\gamma_2\end{bmatrix}(\xi+\lambda_1\tau+\lambda_2,\tau)=\exp(-i\pi\lambda_1^2\tau-2i\pi\lambda_1(\xi+\gamma_2+\lambda_2))\vartheta\begin{bmatrix}\gamma_1+\lambda_1\\\gamma_2+\lambda_2\end{bmatrix}(\xi,\tau). \quad (A.4)$$

Considering the Jacobi $\Theta$ function:

$$\Theta_p(z)=(z;p)_\infty\,(pz^{-1};p)_\infty\,(p;p)_\infty, \quad (A.5)$$

where the infinite $q$-Pochhammer symbols are defined by:

$$(z;p_1,\dots,p_m)_\infty=\prod_{n_i\ge 0}(1-zp_1^{n_1}\dots p_m^{n_m}), \quad (A.6)$$

the Jacobi theta functions with rational characteristics $(\gamma_1,\gamma_2)\in\frac{1}{N}\mathbb{Z}\times\frac{1}{N}\mathbb{Z}$ can be expressed in terms of the Jacobi $\Theta$ functions as:

$$\vartheta\begin{bmatrix}\gamma_1\\\gamma_2\end{bmatrix}(\xi,\tau)=(-1)^{2\gamma_1\gamma_2}p^{\frac{1}{2}\gamma_1^2}z^{2\gamma_1}\,\Theta_p(-e^{2i\pi\gamma_2}p^{\gamma_1+\frac{1}{2}}z^2), \quad (A.7)$$

where $p=e^{2i\pi\tau}$ and $z=e^{i\pi\xi}$.
It is easy to show that the $\Theta_{a^2}(z)$ function enjoys the following properties:

$$\Theta_{a^2}(a^2z)=\Theta_{a^2}(z^{-1})=-\frac{\Theta_{a^2}(z)}{z}\quad\text{and}\quad\Theta_{a^2}(az)=\Theta_{a^2}(az^{-1}), \quad (A.8)$$

$$\prod_{i=0}^{N-1}\Theta_{a^{2N}}(a^{2i}z)=\frac{\left((a^{2N};a^{2N})_\infty\right)^N}{(a^2;a^2)_\infty}\Theta_{a^2}(z). \quad (A.9)$$

# B  Definition of the $N$-elliptic $R$-matrix

The starting point for the definition of the elliptic quantum algebras of vertex type is the $N$-elliptic $R$-matrix in $\mathrm{End}(\mathbb{C}^N)\otimes\mathrm{End}(\mathbb{C}^N)$ associated to the $\mathbb{Z}_N$-vertex model [17–19], given by

$$\mathcal{Z}(z,q,p)=z^{2/N-2}\frac{1}{\kappa(z^2)}\frac{\vartheta\begin{bmatrix}\frac{1}{2}\\\frac{1}{2}\end{bmatrix}(\zeta,\tau)}{\vartheta\begin{bmatrix}\frac{1}{2}\\\frac{1}{2}\end{bmatrix}(\xi+\zeta,\tau)}\sum_{(\alpha_1,\alpha_2)\in\mathbb{Z}_N\times\mathbb{Z}_N}W_{(\alpha_1,\alpha_2)}(\xi,\zeta,\tau)\,I_{(\alpha_1,\alpha_2)}\otimes I_{(\alpha_1,\alpha_2)}^{-1},$$

$$(B.1)$$

where the variables $z,q,p$ are related to the variables $\xi,\zeta,\tau$ by

$$z=e^{i\pi\xi},\qquad q=e^{i\pi\zeta},\qquad p=e^{2i\pi\tau}. \quad (B.2)$$

We set

$$R(z,q,p)=(g^{\frac{1}{2}}\otimes g^{\frac{1}{2}})\mathcal{Z}(z,q,p)(g^{-\frac{1}{2}}\otimes g^{-\frac{1}{2}}), \quad (B.3)$$

where the $N\times N$ matrix $g$ is defined by

$$g_{ij}=\omega^i\delta_{ij},\qquad 1\le i,j\le N\quad\text{with}\quad\omega=e^{2i\pi/N}. \quad (B.4)$$

The normalization factor is chosen as follows:

$$\frac{1}{\kappa(z^2)}=\frac{(q^{2N}z^{-2};p,q^{2N})_\infty(q^2z^2;p,q^{2N})_\infty(pz^{-2};p,q^{2N})_\infty(pq^{2N-2}z^2;p,q^{2N})_\infty}{(q^{2N}z^2;p,q^{2N})_\infty(q^2z^{-2};p,q^{2N})_\infty(pz^2;p,q^{2N})_\infty(pq^{2N-2}z^{-2};p,q^{2N})_\infty}. \quad (B.5)$$

The Jacobi theta functions and the infinite products are defined in Appendix A.
The functions $W_{(\alpha_1,\alpha_2)}$ are given by

$$W_{(\alpha_1,\alpha_2)}(\xi,\zeta,\tau) = \frac{\vartheta\begin{bmatrix} \frac{1}{2}+\alpha_1/N \\ \frac{1}{2}+\alpha_2/N \end{bmatrix}(\xi+\zeta/N,\tau)}{N\vartheta\begin{bmatrix} \frac{1}{2}+\alpha_1/N \\ \frac{1}{2}+\alpha_2/N \end{bmatrix}(\zeta/N,\tau)}\,, \tag{B.6}$$

and $I_{(\alpha_1,\alpha_2)} = g^{\alpha_2} h^{\alpha_1}$ where the $N \times N$ matrix $h$ is such that

$$h_{ij} = \delta_{i+1,j}, \qquad 1 \le i,j \le N, \tag{B.7}$$

the addition of indices being understood modulo $N$.

We recall the following proposition [9], see also [20, 28]:

**Proposition B.1** *The matrix $R(z) \equiv R(z,q,p)$ satisfies the following properties:*
*– Yang–Baxter equation (also holds for $\widehat{R}$):*

$$R_{12}(z)R_{13}(w)R_{23}(w/z) = R_{23}(w/z)R_{13}(w)R_{12}(z), \tag{B.8}$$

*– Unitarity:*

$$R_{12}(z)R_{21}(z^{-1}) = 1\,, \tag{B.9}$$

*– Regularity ($P_{12}$ is the permutation matrix):*

$$R_{12}(1) = P_{12}\,, \tag{B.10}$$

*– Crossing-symmetry:*

$$R_{12}(z)^{t_2} R_{21}(z^{-1}q^{-N})^{t_2} = 1\,, \tag{B.11}$$

*– Antisymmetry:*

$$R_{12}(-z) = \omega\,(g^{-1} \otimes \mathbb{I})R_{12}(z)(g \otimes \mathbb{I})\,, \tag{B.12}$$

*– Quasi-periodicity:*

$$\widehat{R}_{12}(-zp^{\frac{1}{2}}) = (g^{\frac{1}{2}}hg^{\frac{1}{2}} \otimes \mathbb{I})^{-1}\widehat{R}_{21}(z^{-1})^{-1}(g^{\frac{1}{2}}hg^{\frac{1}{2}} \otimes \mathbb{I})\,, \tag{B.13}$$

*where*

$$\widehat{R}_{12}(z) \equiv \widehat{R}_{12}(z,q,p) = \tau_N(q^{\frac{1}{2}}z^{-1})R_{12}(z,q,p)\,, \tag{B.14}$$

*the function $\tau_N(z)$ being defined by*

$$\tau_N(z) = z^{\frac{2}{N}-2}\frac{\Theta_{q^{2N}}(qz^2)}{\Theta_{q^{2N}}(qz^{-2})}\,. \tag{B.15}$$

*The function $\tau_N(z)$ is $q^N$-periodic, $\tau_N(q^N z) = \tau_N(z)$, and satisfies $\tau_N(z^{-1}) = \tau_N(z)^{-1}$.*

**Remark B.1** The crossing-symmetry and the unitarity properties of $R_{12}$ allow to exchange the inversion and the transposition when applied to the matrix $R_{12}$ (or to the matrix $\widehat{R}_{12}$). It provides a *crossing-unitarity relation* (also valid for $\widehat{R}$ thanks to the $q^N$-periodicity of the function $\tau_N$):

$$\left(R_{12}(x)^{t_2}\right)^{-1} = \left(R_{12}(q^N x)^{-1}\right)^{t_2}. \tag{B.16}$$

Note also that the unitarity property for $\widehat{R}_{12}$ now reads

$$\widehat{R}_{12}(z)\widehat{R}_{21}(z^{-1}) = \tau_N(q^{\frac{1}{2}}z)\,\tau_N(q^{\frac{1}{2}}z^{-1}) \equiv \mathcal{U}(z). \tag{B.17}$$

The function $\mathcal{U}(z)$ is extensively used in our discussion. In term of Jacobi $\Theta$ functions, it reads

$$\mathcal{U}(z) = q^{\frac{2}{N}-2}\frac{\Theta_{q^{2N}}(q^2z^2)\,\Theta_{q^{2N}}(q^2z^{-2})}{\Theta_{q^{2N}}(z^2)\Theta_{q^{2N}}(z^{-2})}\,. \tag{B.18}$$

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
