# Peer review of "Elliptic deformation of $\mathcal{W}_N$-algebras"

_SciPost Physics, doi:SciPost Phys. 6, 054 (2019)_

## Round 1 · Referee Report · Anonymous (Referee 1) · 2019-3-15

Strengths

1.This paper develops an approach to a construction of q-deformations of quantum $W_N$ algebras based on the Lax operator of $A_{q,p}({\hat{gl}}(N)_c)$. 2. The main result is a derivation for exchange relations for generators satisfied on a special surface in the space of parameters. 3. Authors compare their construction with a previously known approach to construct quantum $W$-algebra generators. 4. They also analyze a critical level case in details.

Weaknesses

  1. Section 5 contains a lot of technical details which could be moved to another appendix.
  2. A notation $\hat R_{12}(z)$ in (2.3) is not the best one. It can be easily confused with the standard notation $\hat R_{12}(z)=P_{12}R_{12}(z)$. At least, formulas (B.14-B.15) should be quoted immediately after (2.3).

Report

I have read the article "Elliptic deformation of $W_N$ algebras" by J.Avan et al. The paper addresses important problems in the theory of quantum elliptic algebras. It relies on several previous results by the authors and is quite challenging technically but this is probably due to the very nature of the subject.
The main result is the Theorem 3.1 where exchange relations for quantum generators are derived provided that parameters $q,p,c$ lie on a special surface. The generators are constructed in terms L-operators for the $A_{q,p}({\hat{gl}}(N)_c)$ algebra. Then the authors match their construction to the construction derived by Feigin and Frenkel and discuss their differences.
In Chapter 4 they introduce Abelian subalgebras with the required restrictions on parameters and derive Poisson structures on such subalgebras. Section 5 is dedicated to the case of critical level where they present in Proposition 5.2 structural constants of a closed Poisson algebra for the algebra generators.
Section 5 exlpains a reduction to $U_q(\hat{gl}_N)$ at ctirical level.
Overall, the paper is well written and opens some interesting new directions for a future research.
As noted in Conclusion, explicit realizations of these algebras are required similar to the case of
$W_{q,p}(A_N)$ algebras. A comparison with other approaches to elliptis $W$-algebras is also would be desirable.

Requested changes

Overall the paper is well written and I recommend it for publication in the current form.
I notices a couple of points above which could improve the paper but they are not compulsory.

---

## Round 1 · Referee Report · Anonymous (Referee 2) · 2019-4-11

Strengths

The paper clarifies the q-deformation of W_N algebras by relating them to well known quadratic algebra structures. The quite complicated exchange relations follow from more fundamental relations following directly from RLL type relations. It is a rather elegant approach and clearly relates q-Virasoro to elliptic quantum algebras.

Weaknesses

It is written in the continuation of other papers. Although each paper can be read independently, at some point it would be interesting to have a synthetic paper gathering the main results.
The algebra looks very nice, but its practical use for physical applications is elusive at that point. Are the t^k_mn generators useful for some physical application ?. Could they be useful in connection with the spectrum of the Ruisjenaars-Schneider chains ?.
The algebra is much wider than the W_N (or Virasoro) algebra and contains a lot more generators, but no interpretation of these generators is given. Do they have a geometric interpretation in the classical limit? Are they connected in some ways to the bigger algebras such as Ding Iohara Miki?

Report

The paper is one among a long series of works (in particular 17) by the authors to unravel the connections between q-W algebras and elliptic algebras of QISM. More specifically, in this paper, the authors are able using their approach to recover the quadratic relations of the W algebra obtained by Feigin and Frenkel. This is an interesting progress.

Requested changes

It would be nice if the authors clarify the relation between their paper and ref [2]. I can guess that t^j_2,-1 is related to F.F. T_j but the authors should make it clear. At the top of page 5, the t^k_mn are claimed to be defined in [2], could the authors be more precise.

---

## Round 2 · List of Changes

Dear Editor,

We thank the referees for their valuable comments and suggestions. We have done some modifications in the papier according to the recommendations. More precisely: - the notation has been fixed after eq. (2.3) (but the standard notation for P_12 R_12 is with the check, not with the hat). - the algebras that we construct are not much wider than the W_N algebra. Indeed, the generators t_{mn}^{(k)}(z) are defined for a given pair of integers m and n defining the surface (hence the relation between p and q for a given central charge c). So to be clear, we have added a sentence just before corollary 3.2. - the first sentence of section 3.2 could be ambiguous. We made the statement more explicit and rephrased the beginning of this section. We also added a sentence before Remark 3.3. We hope that this clarification answers to the requested changes of referee 2. - we added a paragraph at the end of the conclusion ("A structural issue ... algebra [24]"). This paragraph is related to the point raised by the second referee in the weaknesses of the paper. Accordingly, we added two references (23 and 24 in the present version). We added also a sentence at the end of the second paragraph of the conclusion regarding possible physical interpretations of this algebra. - we modified formula (5.22) (there was a mistake in this formula). - finally, it does not seem appropriate to us to split the section 5 with the details moved to another appendix as suggested by the first referee, since the proof is not only technical and should be read as a whole. - some references were not properly ordered, now it is fixed.

Sincerely yours, The authors

---

## Editorial Decision

published